# The Roles of Calcineurin B-like Proteins in Plants under Salt Stress

**DOI:** 10.3390/ijms242316958

**Published:** 2023-11-30

**Authors:** Oluwaseyi Setonji Hunpatin, Guang Yuan, Tongjia Nong, Chuhan Shi, Xue Wu, Haobao Liu, Yang Ning, Qian Wang

**Affiliations:** 1Tobacco Research Institute, Chinese Academy of Agricultural Sciences, Qingdao 266101, China; seyipatin@gmail.com (O.S.H.); yuanguang1995@163.com (G.Y.); ntj16726@163.com (T.N.); shichuhanycs@163.com (C.S.); 82101215490@caas.cn (X.W.); liuhaobao@caas.cn (H.L.); 2Graduate School of Chinese Academy of Agricultural Sciences, Beijing 100081, China

**Keywords:** salinity, salt stress, calcineurin B-like proteins, CBL-interacting protein kinases, salt tolerance

## Abstract

Salinity stands as a significant environmental stressor, severely impacting crop productivity. Plants exposed to salt stress undergo physiological alterations that influence their growth and development. Meanwhile, plants have also evolved mechanisms to endure the detrimental effects of salinity-induced salt stress. Within plants, Calcineurin B-like (CBL) proteins act as vital Ca^2+^ sensors, binding to Ca^2+^ and subsequently transmitting signals to downstream response pathways. CBLs engage with CBL-interacting protein kinases (CIPKs), forming complexes that regulate a multitude of plant growth and developmental processes, notably ion homeostasis in response to salinity conditions. This review introduces the repercussions of salt stress, including osmotic stress, diminished photosynthesis, and oxidative damage. It also explores how CBLs modulate the response to salt stress in plants, outlining the functions of the CBL-CIPK modules involved. Comprehending the mechanisms through which CBL proteins mediate salt tolerance can accelerate the development of cultivars resistant to salinity.

## 1. Introduction

Salinity is a severe factor affecting the yield of crops [1]. The buildup of excess soluble salts like Na^+^ and Cl^−^ in the soil is what causes salinity [2]. Saline soil can be practically identified when the electrical conductivity (EC) of the soil sample from the selected area is greater than 4 dSm^−1^ and the exchangeable Na^+^ concentration is 15% [3]. At this point, the soil has a NaCl concentration of around 40 mM. According to the Food and Agricultural Organization (FAO), industrialization, excessive fertilization, increased use of irrigation water of poor quality, soil salinization, and natural causes like salt intrusion in coastal zones as a result of rising sea levels are all contributing factors that will cause salinity to affect about 20% of agricultural farmland significantly more in the coming years [4]. There has been an increase in the emigration of farmers from coastal areas [5], which is proof that soil salinity has far-reaching effects, not only on plants but also on humans who benefit directly or indirectly from plants’ productivity. Salt stress affects plants in several ways, and chief among them is a decline in growth and development [6,7]. Researchers are seeking ways to alleviate the impact caused by salt.

Being stationary organisms, plants have evolved mechanisms to sense environmental cues such as salt stress and respond accordingly for adaptation. Perception of these stimuli always triggers the creation of temporary changes in cytoplasmic calcium ion concentration, known as Ca^2+^ signatures [8,9]. This Ca^2+^ signature plays a crucial role in the signaling transduction processes of stress tolerance in plants. Various proteins, such as Calcium-dependent protein kinases (CDPKs), Calmodulin-like proteins (CMLs), Calmodulins (CAMs), and Calcineurin B-like proteins (CBLs), serve as calcium sensors that recognize Ca^2+^ signatures in plant cells and finally trigger transcriptional and metabolic responses to these stresses by modifying the downstream protein targets [10,11]. CBLs always engage with CBL-interacting protein kinases (CIPKs), and the resulting CBL-CIPK complexes exert regulation over a multitude of plant growth and developmental processes, including the regulation of ion homeostasis in response to different stresses, including salinity.

## 2. Effects of Salt Stress in Plants

The effects of salt stress in plants can be detrimental and manifest at multiple levels, from molecular and physiological to morphological and ecological.

### 2.1. Phenotypes of Salt Stress in Plants

Salt stress hinders cell expansion and division in plants, resulting in diminished growth and overall biomass [12,13]. Elevated salt levels induce leaf chlorosis, causing the leaves to be yellow or brown due to disrupted chlorophyll synthesis, alongside leaf withering and abscission [14]. Plants undergoing salt stress often display modifications in root architecture, such as reduced root length, fewer lateral roots, and increased root diameter, aiming to enhance water and nutrient absorption from the soil [15,16]. Salt stress also changes the gravitropism of the root system [17]. Reproductive development is adversely impacted by salt stress, leading to a reduction in flowering, fruit set, and seed yield [18,19].

### 2.2. Physiological Effects of Salt Stress in Plants

Salt stress causes osmotic and ionic stresses which are the consequences of limited water uptake by plants growing in saline soil [20] (Figure 1).

Osmotic stress: The buildup of salts in the soil increases the osmotic potential of the soil solution. This means that the water potential in the soil is reduced, making it more difficult for plants to take up water from the soil [21]. Osmotic potential is a measure of the ability of a solution to draw water into itself [22]. In a saline environment, the osmotic potential of the soil solution is less than that of the plant cells. As a result, water from the plant cells moves towards the soil, causing dehydration and reducing turgor pressure within the plant cells. Due to the higher osmotic potential in the soil (lower water potential), water is less available to the plant roots. Plants need to exert more energy to absorb water against this osmotic gradient. The reduced water uptake leads to the dehydration of plant cells. Dehydration affects various cellular processes, including photosynthesis, enzyme activity, and metabolism [23,24]. It also disrupts the normal expansion necessary for growth and development. The effects of a water deficit in plants include diminished cell turgor and reduced water usage efficiency [8]. Turgor pressure refers to the pressure exerted by the contents of a cell against its surrounding cell wall. When cells lose water due to reduced water uptake, turgor pressure decreases.

Ionic stress: Ionic stress occurs as a result of the accumulation of salts, particularly Na^+^ and Cl^−^, in the plant tissue [25]. This disrupts the normal ionic balance and homeostasis within the plant cells, leading to various physiological and metabolic disturbances. Plants absorb water and essential nutrients, including ions, from the soil through their roots. Under normal conditions, plants maintain a delicate balance of ions, such as K^+^, Ca^2+^, and Mg^2+^, to ensure proper cellular functions and water uptake. However, in a saline environment, the concentration of Na^+^ and Cl^−^ increases, upsetting the balance. Elevated Na^+^ levels can cause toxicity in plants [26] and can hinder critical processes like enzyme activation, photosynthesis, and osmotic regulation [20,21,24].

Oxidative stress: Reactive oxygen species (ROS) buildup brought on by salt stress leads to oxidative damage in plants and causes oxidative stress [27]. In plants, ROS are manufactured in organelles like peroxisomes, chloroplasts, mitochondria, and the apoplast [28]. Leakage of electrons from the electron transport chain (ETC) during salt stress causes the production of mitochondrial ROS, which can then be transformed to H_2_O_2_ by Manganese Superoxide Dismutase Mn-SOD [29]. ROS are generated in peroxisomes as a result of increased photorespiration during salt stress and reduced photosynthetic activity in the chloroplasts during salt stress leads to the formation of ROS [30,31].

## 3. Plant Response to Salinity

Plants have also developed diverse adaptive mechanisms to respond to salinity stress, including osmotic adjustment, ion exclusion, antioxidant defense systems, and morphological adaptations.

### 3.1. Physiological Responses and Adaptations of Plants to Salt Stress

Plants have developed a variety of physiological responses and adaptations to mitigate the harmful impact of salt stress. They regulate stomatal conductance to reduce water loss during salt stress [32], which in turn lowers transpiration and helps in water conservation. Plants often respond to salt stress by closing their stomata, which helps minimize water loss but also reduces carbon dioxide uptake, which is essential for photosynthesis [21]. Additionally, plants utilize specific mechanisms to selectively uptake ions, such as Na^+^, and transport them to specific tissues, like older leaves or vacuoles, to prevent their accumulation in metabolically active tissues and maintain ion homeostasis [33].

To uphold cellular turgor and osmotic balance, plants accumulate compatible solutes [34,35]. These solutes serve as osmoprotectants, shielding proteins and cellular structures from the harmful impacts of increased salt concentrations. Plants may also undergo metabolic adjustments to effectively cope with salt stress [36], such as modifying enzyme activities, energy metabolism, and carbon partitioning to optimize resource utilization during adverse conditions.

### 3.2. Molecular Responses and Adaptations of Plants to Salt Stress

Plants undergo alternative splicing and post-translational modifications of proteins to adapt to salt stress [37,38]. These modifications modulate protein functionality, stability, and subcellular localization, enabling plants to fine-tune their response to salt-induced changes. Additionally, epigenetic alterations such as DNA methylation, changes in histone structure, and small RNA regulation play a role in influencing gene expression and stress responses [39]. The salt stress conditions can impact the epigenetic landscape of plants, potentially affecting gene expression and contributing to their adaptation to stress. In response to salt stress, plants increase the expression of genes linked to stress responses [40,41], ion transporters, osmotic regulation, and antioxidant defense systems. These genes collectively play a crucial role in enhancing salt tolerance by aiding in ion balance, osmotic adjustment, and ROS detoxification.

## 4. CBLs Function as Calcium Sensors in Plants

CBLs are a kind of calcium sensor exclusively found in plants and are upregulated in response to multiple environmental stresses [42]. Elongation factor (EF) hand domains serve as a distinctive feature in Ca^2+^ sensors, including proteins like calcineurin, calmodulins, and CBL. These EF-hands, with a typical helix-loop-helix secondary structure, play a crucial role in Ca^2+^ binding (Figure 2). Notably, the EF-hand domains in CBLs exhibit marked differences compared to Ca^2+^ binding proteins such as CAMs [43]. The first EF-hand domain in CBLs consists of 14 amino acids, contrasting with the 12 amino acids found in other Ca^2+^ sensors [44].

CBL proteins comprise four EF-hands, each characterized by a conserved α-helix-loop-α-helix structure that facilitates Ca^2+^ binding [45,46]. These EF-hands are strategically positioned at fixed intervals, covering distances of 22, 25, and 32 amino acids from EF1 to EF4, respectively [41,43]. The loop region displays a consensus sequence of 12 residues: DKDGDGKIDFEE [45,47]. Positions 1(X), 3(Y), 5(Z), 7(-X), 9(-Y), and 12(-Z) within this sequence play a key role in coordinating Ca^2+^ binding [45,46]. Variations in the amino acids at these specific positions can impact the affinity for Ca^2+^ binding [45,48]. It is noteworthy that EF1 in CBLs features a two-amino acid insertion between position X and position Y [42,46]. Interestingly, some CBLs have been reported to possess three EF-hands [49]. To date, *Arabidopsis thaliana*, a model plant, has revealed the presence of ten (10) CBLs [50,51]. 

CBLs do not act alone, but on binding to Ca^2+^, they interact with a kind of kinase named CIPK. CIPKs phosphorylate the C-terminal region of the CBL protein, which contains the FSPF. Different CBLs can also interact independently with one CIPK. For example, *AtCBL1* and *AtCBL9* can interact with CIPK23, thereby phosphorylating AKT1 to promote K^+^ uptake under low-K^+^ conditions [52,53,54]. CBLs specifically target the plant-exclusive CIPKs family of serine-threonine kinases, characterized by an N-terminal kinase catalytic domain, a C-terminal inhibitory domain, and an NAF motif known as the FISL motif. The N-terminal kinase features a proposed activation loop with conserved serine, threonine, and tyrosine residues. CBLs bind to the FISL motif, followed by a conserved protein phosphatase interaction (PPI), inducing CIPKs to engage with protein phosphatases 2C [55,56,57,58,59,60,61]. The suggested mechanism posits that the interplay between CIPKs, 2C-type protein phosphatases, and downstream target proteins involves phosphorylation and dephosphorylation events [62]. The Salt Overly Sensitive (SOS) pathway (Figure 3) is central to cell signaling in salt stress [63].

Some CBLs also possess myristoylation motifs. Myristoylation is a post-translational modification that is important for the attachment of proteins to membranes [64,65]. The N-terminus of each CBL protein has a characteristic MGCXXSK/T sequence, which is a site for myristoylation. The glycine residue, which is at the second position in this sequence, is the exact point of myristoylation. The addition of a palmytoyl group to the cysteine next to the second glycine increases the affinity of the poorly attached myristoylated CBL proteins to the membrane [65,66].

### 4.1. CBL-CIPK Modules Regulate Ion Homeostasis in Plants under Salt Stress

As discussed earlier in this review (see Section 2.2), salt stress causes ionic stress which disrupts the normal ionic balance and homeostasis within the plant cells, leading to various physiological and metabolic disturbances. In order to prevent the cytoplasm from amassing an excessive amount of Na^+^, which is highly toxic to many essential metabolic processes, plants under salt stress employ a variety of defense mechanisms [67]. By preventing the entry of Na^+^ cells, storing Na^+^ in vacuoles, or secreting Na^+^ through membrane transporters, plants can keep their Na^+^ levels low [68]. SOS1 (salt overly sensitive 1), a plasma membrane Na^+^/H^+^ antiporter, is very effective at keeping the cytoplasm’s level of Na^+^ at a lower level [69,70]. One of the complexes that activate SOS1 is CBL4-CIPK24. The complex uses the energy from the proton gradient produced by the H^+^ gradient produced by the H^+^-ATPase to phosphorylate SOS1, which transports Na^+^ out of the cells [71].

CBL4-CIPK24-SOS1 regulates Na^+^ exclusion in roots, while CBL10-CIPK24-SOS1 regulates Na^+^ exclusion in shoots. These mechanisms are triggered by the binding of Ca^2+^ to CBLs, which is brought on by an abrupt increase in cytoplasmic Ca^2+^ concentration in plants under salt stress. The shoots are typically where CBL10 is expressed and, like CBL4, interacts physically with CIPK24 [72]. CBL10-CIPK24 is localized in the tonoplast, in contrast to the CBL4-CIPK24 complex, which is localized in the plasma membrane [72,73] (Figure 4). This indicates that they may have a function in the vacuolar sequestration of Na^+^.

More so, it was reported that CBL10 does not participate in the SOS1 pathway because, when there was a loss of function of SOS1, the Na^+^ content of the mutant was not different from that of the wild-type [72]. This suggests that CBL10 may play a role in the sequestration of Na^+^ in vacuoles rather than the SOS pathway. Some Na^+^/H^+^ exchangers in the tonoplast may use it to mediate the movement of Na^+^ from the cytoplasm to the vacuole [72]. Although, some recent studies suggested that CBL10 may be involved with the SOS pathway [68,74], the in vivo functional relationship between CBL10 and SOS has not been established [75].

Another way in which plants maintain low Na^+^ levels is by increasing the uptake of K^+^. When the K^+^ concentration rises, the Na^+^ concentration falls, and vice versa [76,77]. Plants absorb K^+^ through the action of the transporters AKT1, AKT2, HKT, and KUP. CIPK23 and CBL1/CBL9 interact physically to open the AKT1 potassium channel [78,79,80] while CIPK6 and CBL4 physically interact to activate AKT2 [81,82]. Phosphorylation is a component of the mechanism by which CBL1/CB9-CIPK23 activates AKT1, but it is absent from the CBL4-CIPK6 mechanism. The translocation of AKT2 to the plasma membrane is triggered by additional palmitoylation of CBL4.

### 4.2. CBL-CIPK Modules Alleviate Osmotic Stress in Plants under Salt Stress

Salt stress restricts the intake of water into plant cells, thereby causing dehydration and altering cell turgidity. The abscisic acid (ABA) level in the cell increases and causes stomatal closure which helps to regulate osmotic homeostasis and water balance [35]. This is a crucial mechanism by which plants adapt to salt stress. Studies have demonstrated that CBL-CIPK modules in osmotic stress responses work either independently of or in dependence on ABA [83,84,85,86,87]. Physical interactions between CBL1 and CIPK1, CBL9 and CIPK1, and CBL4 and CIPK6 cause AKT2 to be activated. ABA-independent stress-responsive signal transduction is carried out by CBL1, ABA-dependent stress-responsive signal transduction is carried out by CBL9, and both types of signal transduction are carried out by CBL4 [83,84,88].

CBL1-CIPK1 and CBL9-CIPK1 complexes are localized at the plasma membrane, but the localization of CIPK1 is not limited to the plasma membrane but also the nucleus [85,89]. The increased expression of the stress markers RD29A, KIN1, RD22, and RAB18 in CIPK mutants demonstrated that CIPK1 controls the expression of genes that are responsive to osmotic stress [85,90]. RD29A, KIN1, RD22, and RAB18 are genes associated with stress responses in plants and are often referred to as stress markers because their expression is induced in response to various environmental stressors, such as drought, salinity, and cold. More importantly, CIPK1 regulates osmotic stress in plants by activating transcription factors CBF1/DREB1b and CBF2/DREB1c, and it does this by interacting with CBLs [85,91].

### 4.3. CBL-CIPK Modules Regulate ROS Signaling

It has been reported that reactive oxygen species (ROS) cause an increase in cytosolic Ca^2+^ and also activate Ca^2+^ channels in root and guard cells [92,93,94]. The mechanism of the connection between Ca^2+^ and ROS signaling was described by [95], where they proposed a model in which Ca^2+^ activates Ca^2+^-dependent protein kinases, leading to the activation of respiratory burst oxidase homologs (RBOHs), which are major ROS-producing enzymes in plants [96]. The activated RBOHs form ROS in the apoplast, which causes the release of Ca^2+^ in nearby cells, which in turn activates Ca^2+^-dependent kinases, leading to cell-to-cell propagation of the signal. CBL1/CBL9-CIPK26 activated respiratory burst oxidase homolog protein F (RBOHF) [97] and respiratory burst oxidase homolog protein D (RBOHD) [98]. SiCBL10-SiCIPK6 complex interacted with RBOHD and caused an increase in ROS, which functions in plant immunity [99]. Low potassium levels, which are a result of salt stress in plants, cause a ROS buildup and then Ca^2+^ activation [62]. The specific mechanisms by which CBL-CIPK complexes modulate ROS signaling are yet to be elucidated.

## 5. The CBL Family and Their Functions in Some Plants under Salt Stress

In this review, we outline the roles of CBL family members in specific plants under salt stress as reported in several studies.

CBL1 has been shown to facilitate responses to salt stress in Arabidopsis [83]. Only about 25% of *cbl* mutants survived a 100 mM NaCl treatment, while 85% of wild-type seedlings survived the same NaCl treatment. To determine whether CBL1 mediates salt responses in adult plants, adult plants were treated with a 100 mM NaCl solution, while the control plants were treated with 200 mM mannitol. Mutant plants showed substantial lesions one week after the salt treatment while wild-type plants showed very minimal lesions, indicating that CBL1 functions in salt responses. However, the mutant and wild-type plants did not develop lesions after mannitol treatment, indicating that the phenotype was specifically caused by salt stress [83]. Also, it has been demonstrated that the expression of *CBL1* from *Sedirea japonica* rescued salt and osmotic stress hypersensitivity in *A*. *thaliana* cbl mutant [100]. *Atcbl1* mutant seedlings exhibited more stunted growth and severe leaf chlorosis when subjected to salt and osmotic stress than the wild-type [84]. However, the over-expression of *SjCBL1* in *Atcbl1* mutants reinstated its salt tolerance; hence, the complemented (*cbl1*/*SjCBL1*) plants exhibited better growth than the cbl mutant seedlings in the high salt medium [100].

CBL2 and CBL3 have been demonstrated to interact with CIPK21 in Arabidopsis using yeast two-hybrid analysis and confirmed by vector swapping and the BiFC assay [101]. Coexpression of CBL2 or CBL3 with CIPK21 caused the preferential localization of CIPK21 to the tonoplast, thereby suggesting the mediation of responses to salt stress, and *cipk21* mutants were hypersensitive to high salt and osmotic stress conditions [101].

Expression of *BnaCBL4* (from rape seed) in the cbl4 mutant reduced its hypersensitivity to salt and its overexpression in Arabidopsis enhanced tolerance to salt stress compared to the wild-type [102]. Recently, it was discovered that overexpressing *CsCBL4* in Arabidopsis increases the salt tolerance of the *CBL4* mutant, while silencing *CsCBL4* or *CsCIPK6* in cucumber increases salt sensitivity [82]. It has been revealed that out of the seven CBLs encoded in the genome of the foxtail millet (*Setaria italica*), only *SiCBL4* and *SiCBL5* are involved in salt responses [103]. Overexpression of *SiCBL4* and *SiCBL5* in the *Atsos3-1* mutant enhanced its salt tolerance as their lengths became longer [103].

As stated earlier, the over-expression of *SiCBL5* in foxtail millet increases its salt tolerance by modulating Na^+^ homeostasis [103]. However, overexpression of *NtCBL5A* induced salt super-sensitivity with necrotic lesions of leaves [104]. In this study, wild-type and over-expressing lines were treated under normal and saline conditions. In typical circumstances, no distinctions were observed between the phenotypes of the over-expressing lines and wild-type. However, in saline conditions, the overexpressing leaves showed early signs of chlorosis and developed necrotic lesions within two weeks, and high Na^+^ was primarily responsible for these necrotic lesions. The necrotic lesions may be caused by defective photosystems and accumulation of ROS [104], but further research is needed to confirm this.

It was reported that the ectopic expression of *TdCBL6* from wheat (*Triticum dicoccoides*) in *Arabidopsis thaliana* enhances its salt tolerance [105]. The *TdCBL6* overexpressing lines exhibited lower ion leakage and higher levels of photosynthetic efficiency than wild-type plants under NaCl stress conditions. It was also reported that *CBL7* genes play a role in regulating the response to salt in sugar beets [42]. Transcriptome analysis indicated that *CBL7* gene expression increased significantly in the salt-tolerant cultivar but not in the salt-sensitive cultivar. The salt-tolerant cultivar displayed better growth under salt stress.

*CBL8* was reported to have mediated the salt response in *A. thaliana* under high salt stress [106]. It was observed that *CBL8* mutant seedlings exhibited a reduced survival rate compared with the wild-type when both groups were subjected to 150 mM NaCl. However, this survival rate was restored to wild-type levels in *CBL8*/*CBL8* complementation lines. Similarly, overexpression of *AtCBL8* in *Nicotiana tabacum* enhanced its salt tolerance, as the overexpression lines had significantly higher fresh weight than the wild-type under high salt stress [106]. It was also demonstrated in the same study that CBL8 interacted with CIPK24 to activate SOS1 to extrude Na^+^ under high salinity.

Overexpression of *ThCBL9* (from *Thellungiella halophila*) in *A. thaliana* increased its tolerance to salt [107]. The transgenic lines and wild-type plants were treated with NaCl and the results showed that increasing the concentration of NaCl led to a greater decline in the germination of the wild-type compared to the transgenic lines. Another study revealed that overexpression of *ZmCBL9* (from maize) in the Arabidopsis *CBL9* mutant rescued it from hypersensitivity to salt [108]. Also, a comparison of the salt tolerance of wild-type Arabidopsis and its *CBL9* mutant showed that the mutant growth was slower than that of the wild-type when both were subjected to high salt concentrations [109].

It has been demonstrated that CBL10 from tomatoes (*SlCBL10*) contributes to improved plant growth during salt stress by modulating the balance of Na^+^ and Ca^2+^ [110]. When *Slcbl10* mutant plants were subjected to short-term salt treatments, the aerial parts of both young and adult plants were severely damaged. Vegetative growth was impeded at the young stage and the leaflets exhibited chlorosis and apical collapse. Similarly, the adult plants also showed abnormal growth when subjected to short-term salt treatments. However, wild-type plants grew better than *Slcbl10* mutant plants under the same conditions of salinity [110]. Also, CBL10 has been shown to enhance salt tolerance in *A. thaliana*, by physically interacting with CIPK8 [68]. The CBL10-CIPK8 complex then activated SOS1 to extrude Na^+^ from the plant, thereby relieving it of salt stress. Table 1 provides a summary of the roles played by CBLs in some salt-stressed plants.

## 6. Concluding Remarks and Future Indications

Due to its severe impact on agricultural productivity, it is imperative to conduct extensive research aimed at mitigating the adverse effects of soil salinity and also employ the right management to reduce salinity. The repercussions of salinity on plants encompass osmotic imbalance, ionic toxicity or imbalances (resulting from excessive Na^+^ and Cl^−^ uptake), nutritional deficiencies, and oxidative bursts (primarily caused by free radicals or reactive oxygen species, ROS).

A group of calcium sensors, known as calcineurin B-like proteins (CBLs), bind to Ca^2+^ when the cytoplasm’s calcium levels rise. The interaction between CBLs and CBL-interacting protein kinases (CIPK) activates downstream stress-responsive proteins. Recent studies have comprehensively outlined the roles of CBLs in various plants when subjected to salt stress, demonstrating that each member of the CBL family contributes to salt tolerance in certain plants. Intriguingly, CBL8 has demonstrated the ability to augment salt tolerance during severe salt stress conditions. Conversely, overexpression of *NtCBL5A* resulted in salt hypersensitivity in *N*. *tabacum*.

Multiple studies emphasize the importance of maintaining low Na^+^ levels, alleviating osmotic stress, and regulating ROS signaling in plants to effectively combat the threat of soil salinity. Strategies, such as exporting Na^+^ out of the cell, sequestering Na^+^ in the vacuole, augmenting K^+^ uptake, regulating osmotic balance, and activating transcription factors, are among the mechanisms by which the CBL-CIPK signaling pathways enhance plant tolerance to salinity. Future research endeavors should delve into further elucidating how CBL proteins bolster salt tolerance in plants, and the specific mechanisms by which CBL-CIPK complexes modulate ROS signaling as crop breeders can potentially modify these genes in plants to increase crop yield and meet the demands of the growing global population.

## Figures and Tables

**Figure 1 ijms-24-16958-f001:**
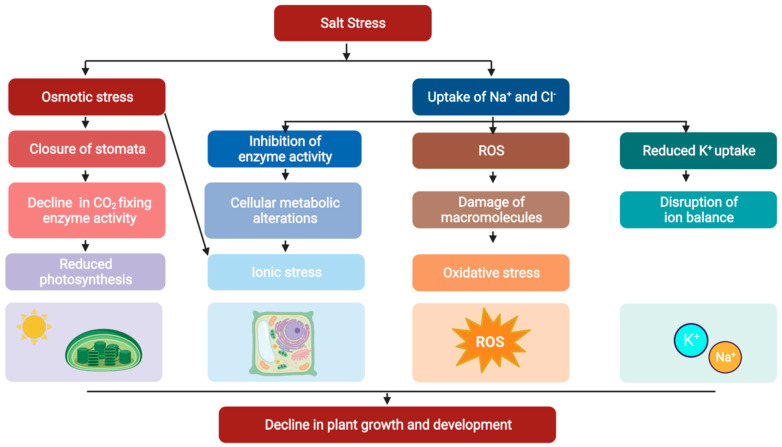
A schematic diagram showing the physiological impacts of salt stress in plants.

**Figure 2 ijms-24-16958-f002:**
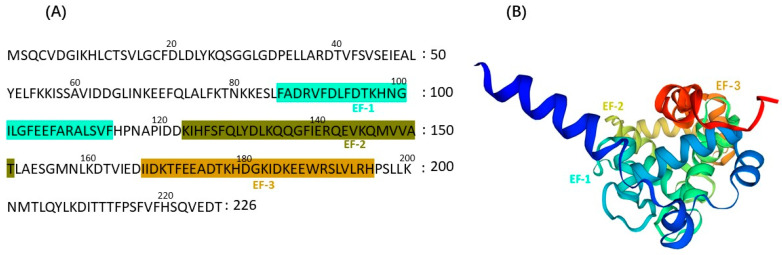
Structures of AtCBL2. (**A**) Amino acid sequence of AtCBL2 and the positions of three EF-hands predicted by SMART (**B**); The 3D structure of AtCBL2 predicted by SWISS-MODEL.

**Figure 3 ijms-24-16958-f003:**
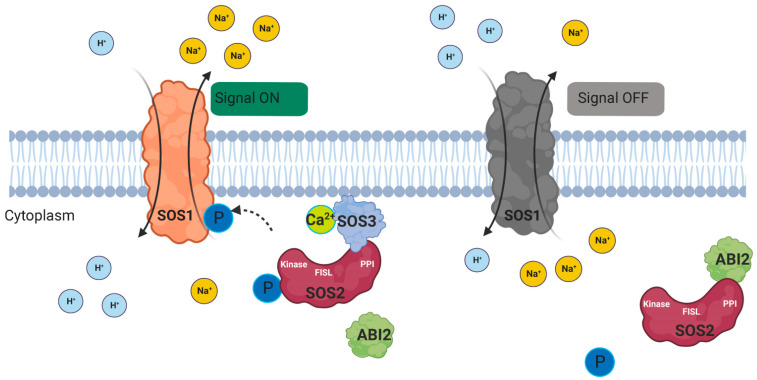
A typical model of the CBL-CIPK pathway (SOS pathway): In response to salt stress, the concentration of cytoplasmic Ca^2+^ increases, leading to the binding of SOS3 to Ca^2+^. This interaction triggers molecular alterations in SOS3, allowing it to physically interact with SOS2 at its NAF/FISL motif. SOS2 undergoes phosphorylation by a kinase, resulting in its activation, and then phosphorylates SOS1. This activation of SOS1 facilitates the efflux of Na^+^ from the cell. Once the stress subsides, ABI2 binds to the PPI motif, leading to the dephosphorylation of both SOS2 and SOS1. (SOS3: CBL4, SOS2: CIPK24, SOS1: Salt overly sensitive 1, ABI2: PPC2 type protein phosphatase.)

**Figure 4 ijms-24-16958-f004:**
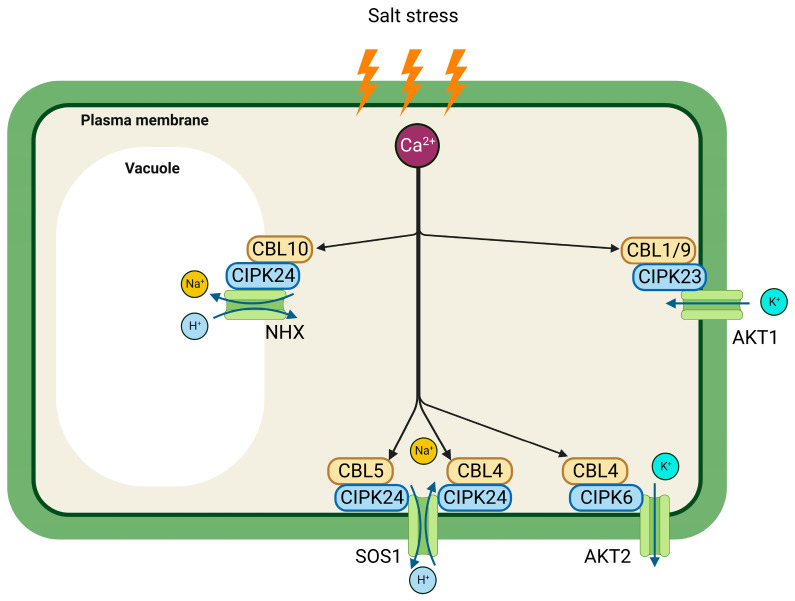
Various CBL-CIPK complexes and their roles in ion homeostasis are shown schematically (SOS1: Salt Overly Sensitive 1, AKT1: Arabidopsis K^+^ Transporter 1, Arabidopsis K^+^ Transporter 2, NHX: Na^+^/H^+^ Exchanger).

**Table 1 ijms-24-16958-t001:** Summary of the functions of the CBL family in named plants.

CBL Proteins	Plant Sources	Function	CBL-CIPK Complex	References
CBL1	*S. Japonica* (Orchid of Nago)	Rescues plant from salt hypersensitivity	CBL1-CIPK1	[100]
	*A*. *thaliana*	Increases salt tolerance	-	[83,84]
CBL2	*A*. *thaliana*	Enhances salt tolerance	CBL2-CIPK21	[101]
CBL3	*A*. *thaliana*	Enhances salt tolerance	CBL3-CIPK21	[101]
CBL4	*Cucumis sativum* (Cucumber)	Enhances salt tolerance	CBL4-CIPK6	[82]
	*Brassica napus*	Enhances salt tolerance	CBL4-CIPK24	[102]
CBL5	*S*. *italica* (Foxtail millet)	Maintains Na^+^ homeostasis and enhances salt tolerance	CBL5-CIPK24	[103]
	*N*. *tabacum* (Common tobacco)	Overexpression causes necrotic lesions	-	[104]
CBL6	*T*. *dicoccoides* (Wheat)	Enhances salt tolerance	-	[105]
CBL7	*Beta vulgaris* (Sugar beet)	Gene expression significantly increased under salt stress	-	[42]
CBL8	*A. thaliana*	Enhances tolerance under high salt stress	CBL8-CIPK24	[106]
CBL9	*T. halophilla*	Increases salt tolerance	CBL9-CIPK23	[107]
	*Zea mays* (Maize)	Rescues plant from salt hypersensitivity	CBL9-CIPK23	[108]
	*A. thaliana*	Increases salt tolerance	CBL9-CIPK23	[109]
CBL10	*Solanum lycopersicum* (Tomato)	Protects growing tissues from salt stress	-	[110]
	*A. thaliana*	Enhances salt stress tolerance	CBL10-CIPK8	[68]

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
