# Peer review of "The Roles of Calcineurin B-like Proteins in Plants under Salt Stress"

_ijms, 2023, doi:10.3390/ijms242316958_

Round 1
Reviewer 1 Report
Comments and Suggestions for Authors
The manuscript is a valuable and interesting review that introduces the reader to the role of CBL proteins in plants under salt stress conditions. Despite its merits, the work needs some improvements.
1. The paper is too broad and contains superfluous information to describe salt stress in general. Thus, the authors should consider removing or shortening and modifying subsections 2.1 (p. 2) and 3 (p. 4). The repetition of certain information in a paper that is primarily intended to describe the role of CBLs in salt stress detracts from the clarity of the information presented.
2. Page 3, l. 95, remove: “tiny pores on the leaf surface”. I think the reader of this article knows what the stomata are. Therefore, this information should be removed from the text.
3. Page 3, l. 122: Add a citation at the end of the sentence.
4. Page 3, l. 202: Is it the subsection title? It is unclear (?).
5. Page 4, l.166-172: The insertion of a figure showing the 3D structure of a selected CBL protein from a plant should be considered.
6. Page 7, Figure 3: I think the abbreviations used in the figure should be explained in the figure caption. The text mentions that CIPK1 is localised in the cell nucleus (p. 7, l. 260). This information can also be marked in Fig. 3.
7. Page 7, l. 261: “stress markers RD29A, KIN1, RD22, and RAB18”. What are the stress markers RD29A, KIN1 etc.? I think this should be explained briefly.
8. Page 7, l. 271, RBOHs are not major ROS-producing enzymes in plants (?). I do not think so. In plant cells, there are stronger generators of ROS (e.g. H2O2), i.e. peroxisomes, chloroplasts and mitochondria (see e.g. older works: Antioxid Redox Signal (2009) 11: 861, Acta Biochim Pol (2007) 54:39).
9. Page 8, l. 285: “CBL family members”. I think it would be a good idea to include a reconstructed phylogenetic tree for selected CBL proteins from plants in the text of the manuscript.
Minor comments
All plant names in Latin should be written in italics.
All gene abbreviations should be written in italics, e.g.: cbl, cbl4, cbl9 etc.
Reviewer 2 Report
Comments and Suggestions for Authors
I appreciate reading the manuscripts. My suggestions are:
- row 56 on or in plants? Be consistent with header.
- rows 67-68Salt stress change also gravitropism of root system. See https://doi.org/10.3390/plants12020412
- Section 3,2, I would add that other than the one reported here there are several genes involved in the plant response please see https://doi.org/10.3390/ijms22126378
- row 162 Would it be possible to add a figure showing the helix-loop?
- row 285-286 This introductory sentence about this review is a bit late, I would simply delete it.
- rows 325-326 species names should bi in italic.
- row 361 other than mitigating salt effect should be mentioned that also right management should be done to reduce salinity.
